# Long-Term Maintenance Planning Method of Rural Roads under Limited Budget: A Case Study of Road Network

Chao Han [1,2,3], Jiuda Huang [2,*], Xu Yang [1], Lili Chen [2] and Tao Chen [2]

1 Highway Institute, Chang'an University, Xi'an 710061, China; hc1527@jsti.com (C.H.); yang.xu@chd.edu.cn (X.Y.)
2 JSTI.GROUP Co., Ltd., Nanjing 211112, China; hjd984@jsti.com (L.C.); cll295@jsti.com or ct985@jsti.com (T.C.)
3 NERC-ARM, National Engineering Research Center of Advanced Road Materials, Nanjing 210019, China
* Correspondence: hjd984@outlook.com; Tel.: +86-15951965112

**Abstract:** At present, the task of maintaining and managing rural roads in China is becoming increasingly severe. To solve the problems of insufficient scheme benefits, complex feasible solutions, and low optimization efficiency in long-term maintenance planning of rural road networks under a limited budget, it is urgent to develop maintenance decision-making models and optimization methods suitable for rural roads in China. This paper focuses on the critical aspects of performance evaluation, prediction, and decision-making. Firstly, this paper proposes evaluation indicators and maintenance countermeasures suitable for rural roads, combining them with the characteristics of rural road performance degradation. Based on different treatment measure levels, RPCI and RRQI performance prediction models are established. On this basis, an improved heuristic optimization method is proposed, which realizes rapid optimization of the most cost-effective solution. Finally, the model and method proposed in this paper are applied to the case analysis of 10 rural roads in Haimen City, generating 171 optimal maintenance sections, further verifying the feasibility and effectiveness of the model. The study provides a theoretical basis for the scientific management of rural road maintenance.

**Keywords:** rural roads; prediction models; improved heuristic methods; maintenance benefits; maintenance strategy

## 1. Introduction

Rural roads in foreign countries are typically referred to as county, township, and village or low-volume roads in China. Rural roads are a critical component of China's road network, essential for rural economic and social development. China began large-scale rural road construction in 2003, with a total length of 4.5314 million kilometers by the end of 2022, accounting for 84.6% of all roads. A developed rural highway transportation network has been established, centered around county towns, with townships as nodes and village groups as the network.

In recent years, rural road development in China has shifted from construction to maintenance due to the rapid growth of the scale of rural road construction. With limited maintenance funds, local road management departments face challenges in managing such a vast mileage of rural roads effectively while improving maintenance fund utilization efficiency and guaranteeing road service levels. The current maintenance and management system applies high-grade road maintenance decision-making without considering differences in pavement structure, traffic volume, and funding budgets, leading to issues like unreasonable pavement performance evaluations and inadequate scientific maintenance decision-making in actual management processes.

Therefore, it is necessary to establish a decision-making method for rural road surface maintenance through systematic research. Selectively locating maintenance sections and measures can effectively maintain a high level of service for rural roads.

Research on low-flow rural road maintenance decision-making has been conducted abroad. Pantha [1] et al. (2010) used Geographic Information System (GIS) to determine the priority map of pavement maintenance in the Himalayas and developed a budget, time, and resource-constrained decision model for rural road maintenance. Mergi [2] et al. (2012) evaluated pavement conditions using the Pavement Condition Index (PCI) and adopted a "worst PCI priority" strategy for Sudan's rural road networks. Mathew [3] et al. (2015) created a dual-objective deterministic optimization model for rural road network maintenance and proposed a constrained genetic algorithm as the optimization tool. Torres-Machi [4] et al. (2017) studied the impact of road environmental performance on road maintenance decision management, integrating technical, economic, and environmental aspects into maintenance plan design, and developed a tool for SUS optimization design.

Agarwa [5] et al. (2017) studied Indian rural roads and found that maintenance decisions primarily depend on functional, structural, and importance conditions, proposing a two-stage optimization process. Pasindu [6] et al. (2020) established a multi-objective decision-making framework based on pavement structure type selection to address funding, technology, and human resource issues in low-traffic rural road maintenance, providing tools for local highway planning and decision-making. Yogesh [7] et al. (2023) researched and developed a long-term rural road network planning method, applied ant colony optimization, and found it to be 14% more efficient than existing models through verification analysis.

Pavement maintenance decision-making has widely employed various methods, from simple ranking to complex mathematical planning. However, each method has limitations in solving real-world optimization problems. For instance, the ranking method is straightforward but lacks holistic conservation planning over the entire period. It struggles with sorting multiple objectives (Kabir [8] et al. 2014, Choi [9] 2015, Abu Dabous [10] et al. 2019). The mathematical planning model considers different sections, schemes, and time series within the road network but suffers from unstable solutions and slow processing for large-scale problems (Fecarotti [11] et al. 2021). Artificial intelligence methods like genetic algorithms, artificial neural networks, and fuzzy set theory have high processing power but are sensitive to environmental factors, require ample sample sizes, learn slowly, and may not yield long-term optimal solutions (Han [12] et al. 2021, Hanandeh [13] et al. 2022, Mohimenul [14] et al. 2023).

To address the issues of inadequate scheme accuracy, complex decision-making, and low optimization efficiency in rural road maintenance with limited budgets, this paper proposes a maintenance decision model suitable for rural roads. The model combines heuristic optimization algorithms and aims to maximize overall benefits throughout the life cycle by considering pavement performance. By establishing a reasonable decision model and optimization algorithm, the scheme can be quickly and accurately optimized. A case study of 10 rural roads in Haimen City, Jiangsu province, with a total road network length of 280 km is conducted to research the application of long-term maintenance planning from 2023 to 2036. The model determines a maintenance strategy with a reasonable budget and best benefits, verifying its feasibility and effectiveness. This scientifically and effectively guides rural road maintenance management in China.

## 2. Research Conditions and Methods

### 2.1. Key Indicators and Conditions

#### 2.1.1. Pavement Performance Evaluation

Considering the characteristics of low road grade (mainly Grade III and IV roads), slow driving speed (not higher than 60 km/h), and small traffic load in rural roads in China, it is often not necessary to select all road performance indicators as evaluation parameters in the actual evaluation of their road conditions (Tang [15] et al. 2021, Shtayat [16] et al.

2020). Therefore, according to the actual road characteristics of the rural road network in the Jiangsu province, this paper ignores the requirements for pavement anti-skid performance and rutting. Two key indicators, namely, the Rural Road Pavement Surface Condition Index (RPCI) and the Rural Road Pavement Riding Quality Index (RRQI), are mainly used for calculation and evaluation, as shown in the following Formulas (1) and (2) and Table 1. In addition, in order to further understand the internal conditions of rural roads, this paper also supplements the evaluation of the internal health status of the pavement based on the RIPCI auxiliary index based on bottom-finding radar, as shown in Equation (3) (this index is not included in the comprehensive evaluation as an influencing factor to evaluate the road condition attenuation).

$$RPCI = 100 - a_0 CR^{a_1} - \sum W_i N_i \tag{1}$$

$$RRQI = \frac{100}{1 + a_2 e^{a_3 IRI}} \tag{2}$$

$$RIPCI = 100 - 15 * IDI^{0.412} \tag{3}$$

where *CR* is Pavement Cracking Ratio; *IRI* is International Roughness Index; *IDI* is the Inner Pavement Distress Ratio; $a_0$, $a_1$, $a_2$, and $a_3$ are related calculation parameters; *Ni* is the number of different types of damage within the surveyed road section; $W_i$ is a unit deduction for different types of damage.

**Table 1.** Key evaluation indicators and grading standards for pavement performance.

| Evaluating Indicator | | Evaluation Grade | | | |
| --- | --- | --- | --- | --- | --- |
| | | Excellent | Good | Average | Inferior and Poor |
| Pavement damage | RPCI | ≥90 | ≥80 | ≥70, <80 | <70 |
| | CR (%) | ≤0.4 | ≤2.6, <0.4 | >2.6, ≤7.8 | >7.8 |
| Pavement driving quality | RRQI | ≥90 | ≥80, <90 | ≥70, <80 | <70 |
| | IR (m/km) | ≤3.0 | >3.0, ≤4.7 | >4.7, ≤5.8 | >5.8 |

### 2.1.2. Maintenance Measures and Effects

This paper recommends the selection of typical maintenance technologies suitable for rural roads, given their large maintenance scale, complex road shape, low maintenance funds, and low level of mechanization. The recommended measures include five strength grades: P1 (slurry sealing layer), P2 (thin layer cover), P3 (crushed stone regeneration: using a multi hammer crusher to break the old road surface into smaller particle sizes, compacting it with a roller, and then adding a new road structure), P4 (milling and repaving one layer), and P5 (milling and repaving two layers). By calculating the costs of each different maintenance technology and evaluating previous engineering application effects, this paper provides the cost and improvement in pavement damage (RPCI) and driving quality (RRQI) after implementation, as shown in Table 2.

**Table 2.** Maintenance technology cost and effect after implementation.

| Measures | Cost (China RMB) ¥/m² | Life/ Year | Implementation Effect (Improvement Value) | |
| --- | --- | --- | --- | --- |
| | | | RPCI | RRQI |
| P1: slurry sealing layer | 29 | 2–4 | MIN (100, RPCI + 10) | MIN (100, RRQI + 10) |
| P2: thin layer cover | 90 | 4–6 | MIN (100, RPCI + 10) | MIN (100, RRQI + 10) |
| P3: crushed stone regeneration | 72 | 4–6 | MIN (100, RPCI + 20) | MIN (100, RRQI + 20) |
| P4: milling and repaving 1 layer | 134 | 6—8 | 100 | 100 |
| P5: milling and repaving 2 layers | 298 | 7–9 | 100 | 100 |

### 2.1.3. Treatment Conditions

When conducting multi-year maintenance planning for rural roads, multiple treatment strategies can be applied for one pavement performance state. This paper uses a decision tree model to analyze and determine the maintenance strategies for county and township roads, selecting RPCI and RRQI as key decision indicators and different strength level measures as treatment plans. By setting reasonable treatment thresholds, decision tree models for the maintenance of asphalt and cement pavements in different combination states are established, as shown in Figures 1 and 2.

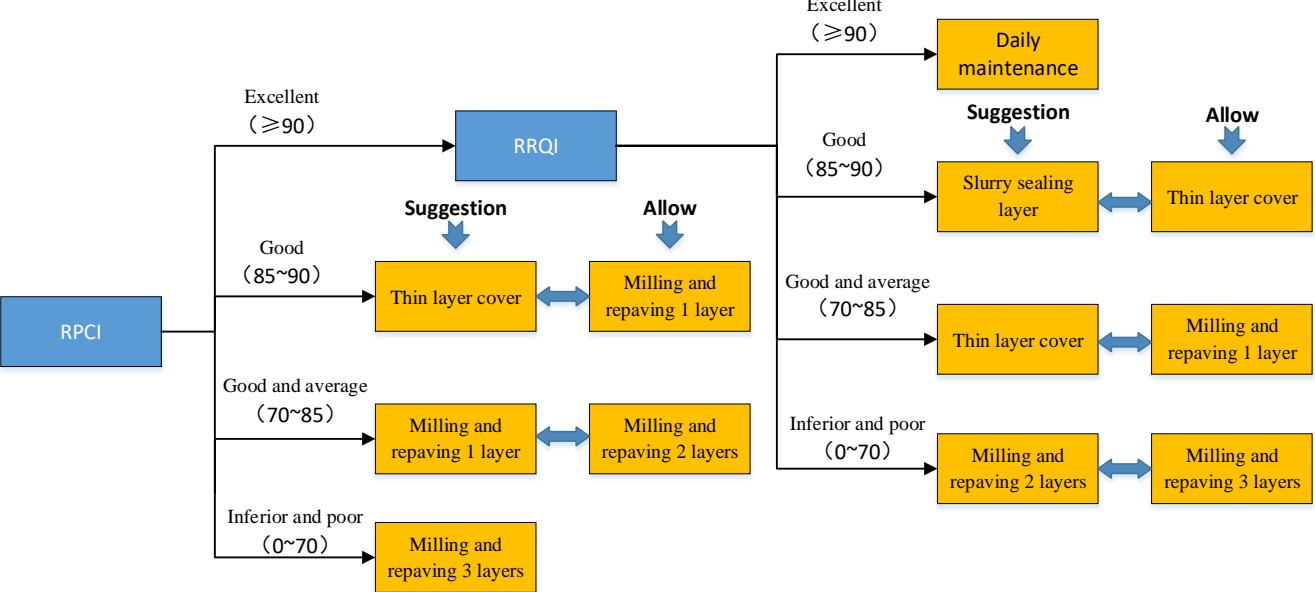

**Figure 1.** Decision tree for rural road asphalt pavement maintenance.

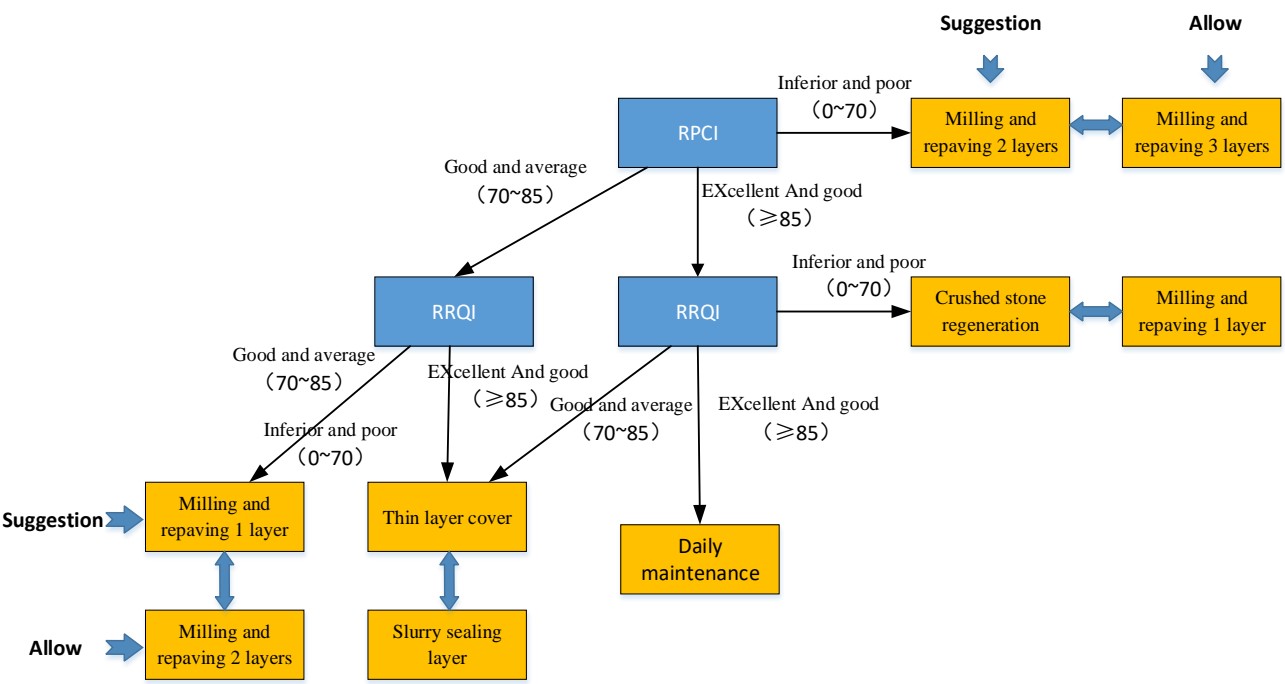

**Figure 2.** Decision tree for cement pavement maintenance of rural roads.

*2.2. Analysis of Pavement Performance Development Law*

2.2.1. Performance Decay Analysis

According to statistical analysis of performance testing data during the life cycle of rural roads in China and relevant literature surveys, road performance shows different attenuation trends when different maintenance opportunities and measures are selected (Yu.J [17] et al. 2015, Stein [18] et al. 2018). Figure 3 summarizes and presents decay curves of road performance under three different treatment strategies: preventive maintenance, functional repair priority, and structural repair priority. The bold curve (a) represents the early use of preventive maintenance techniques such as slurry sealing layer when small pavement damage occurs, resulting in slow performance decay to t1 and corresponding repairs to maintain high road conditions. However, as road age increases, overall pavement performance continues to decline in later stages until the road loses its intended service function. The bold curve (b) represents regular maintenance using techniques like crushed stone regeneration, milling, and repaving one or two layers when the pavement structure is damaged to a certain ex-tent and performance decays to t2 after the early adoption of preventive maintenance strategies to restore it to original technical condition. The bold curve (c) only starts selecting the necessary structural repair measures like milling and repaving three layers for maintenance when pavement performance continues to decline until the severe loss of use function at t3, restoring pavement performance quickly to its original level. This curve has a relatively long degradation period with average overall road performance.

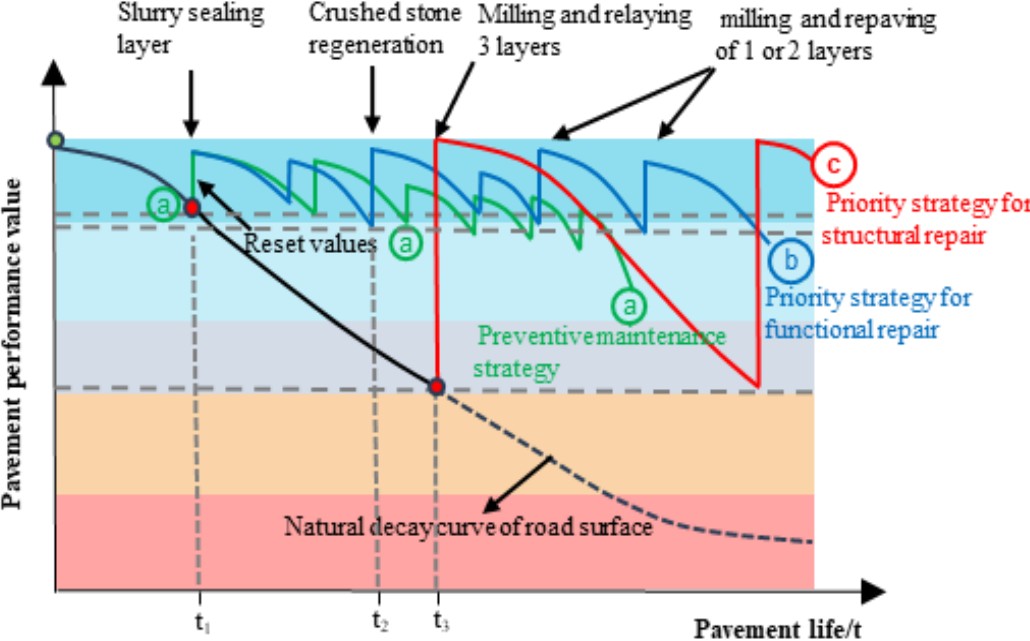

**Figure 3.** Performance decay curve of rural road pavement.

Based on the above analysis, to address the characteristics of road condition degradation during the life cycle of rural roads, selecting a suitable functional repair priority strategy for curve (b) can maintain the overall performance degradation of road conditions within a reasonable range and achieve long-term stability and preservation of rural road pavement structure.

### 2.2.2. Selection of Typical Road Sections

Considering differences in pavement structure, maintenance history, and traffic volume, rural roads in different regions often exhibit varying decay patterns. Therefore, before analyzing the degradation of rural road pavement performance, typical road sections should be selected based on these differences (Pantuso [19] et al. 2019). This paper selects typical rural road networks in Haimen, Guannan, and Yangzhou in Jiangsu province, China through extensive investigations and divides them into asphalt and cement pavement sections based on structural types. Then, typical road sections are chosen for pavement performance analysis according to the maintenance history of various treatment measures, such as slurry sealing layer, thin layer cover, and crushed stone regeneration. The specific selection of typical road sections is shown in Table 3.

**Table 3.** Selection of typical sections of rural roads in the Jiangsu province.

| Pavement Structure | Maintenance History | Road Section Location (Region and Year) | Thickness (cm) | Traffic Level | RIPCI |
|---|---|---|---|---|---|
| Asphalt pavement | Original pavement | K003 + 200-K005 + 300 (Haimen City, 2008) | 30 | Medium | Excellent (94.69) |
| | Slurry sealing layer | K009 + 100-K010 + 000 (Haimen City, 2020) | | | |
| | Thin layer cover | K006 + 700-K007 + 200 (Haimen City, 2015) | | | |
| | Crushed stone regeneration | K015 + 700-K017 + 200 (Haimen City, 2015) | | | |
| | Milling and repaving one layer | K034 + 900-K036 + 600 (Haimen City, 2015) | | | |
| | Milling and repaving two layers | K049 + 100-K053 + 200 (Haimen City, 2015) | | | |
| Cement pavement | Original pavement | K012 + 000-K016 + 2000 (Guannan County, 2008) | 22 | Medium | Excellent (92.02) |
| | Slurry sealing layer | K007 + 500-K009 + 000 (Guannan County, 2019) | | | |
| | Thin layer cover | K010 + 500-K013 + 000 (Guannan County, 2019) | | | |
| | Crushed stone regeneration | K016 + 000-K016 + 600 (Guannan County, 2019) | | | |
| | Milling and repaving one layer | K027 + 200-K028 + 400 (Yangzhou City, 2018) | | | |
| | Milling and repaving two layers | K024 + 100-K026 + 500 (Yangzhou City, 2015) | | | |

### 2.2.3. Determination of Prediction Model

In this section, we aim to establish a reasonable prediction model based on the influence of different treatment measures to evaluate the decay law of pavement conditions with time. In this paper, a large number of detection data of the above-mentioned selected typical sections are analyzed, and combined with the degradation law of rural highway pavement performance, the pavement performance prediction models of rural road RPCI and RRQI based on different treatment measures are established.

(1) Prediction model for RPCI

Based on the regular inspection data of rural roads in Jiangsu province from 2016 to 2021, the RPCI data of typical sections of rural asphalt pavement and cement pavement were fitted and analyzed. The main steps include: (1) grouping the inspection data and determining the corresponding initial decay year; (2) determining the basic parameters through data fitting analysis; (3) considering the impact of the internal health condition index RIPCI on the decay pattern of rural road pavement (RIPCI > 90, the decay coefficient is 1, 80 < RIPCI < 90, the decay coefficient is 0.8, 70 < RIPCI < 80, the decay coefficient is 0.6, RIPCI < 70 decay coefficient is 0); (4) according to the verification of other groups of data, optimize and determine the best parameters when the fitting accuracy requirements are met. Finally, RPCI decay models of five different treatment measures, such as slurry sealing layer, thin layer cover, crushed stone regeneration, milling and repaving one layer, and milling and repaving two layers, are established, respectively, as shown in Figures 4 and 5 below.

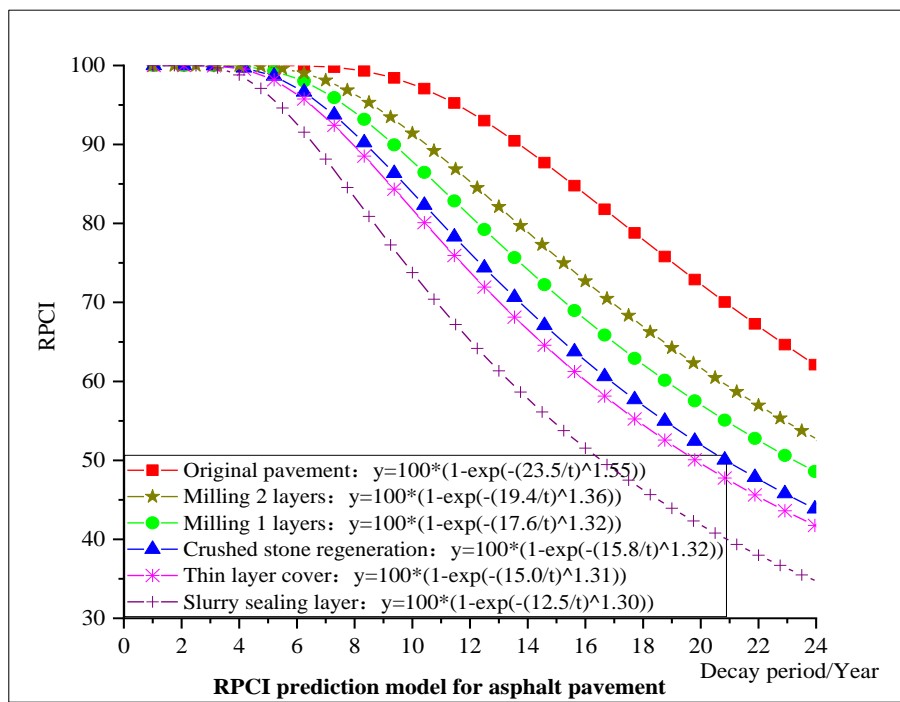

**Figure 4.** RPCI decay curve of asphalt pavement in rural roads over time.

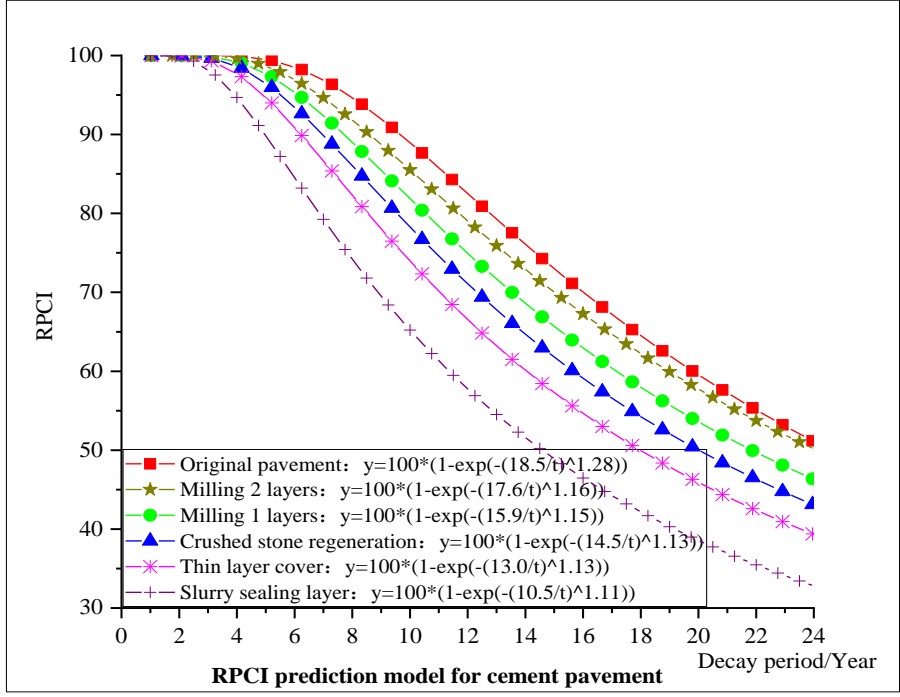

**Figure 5.** RPCI decay curve of cement pavement damage in rural roads over time.

(2)  Prediction model for RRQI

Based on the road condition inspection data from 2016 to 2021, the RRQI data of typical sections of asphalt pavement and cement pavement on rural roads in Jiangsu province were fitted and analyzed. The RRQI decay curves of five different treatment levels were established, including slurry seal layer, thin layer cover layer, crushed stone regeneration, milling and resurfacing one layer, and milling and resurfacing two layers, as shown in Figures 6 and 7 below.

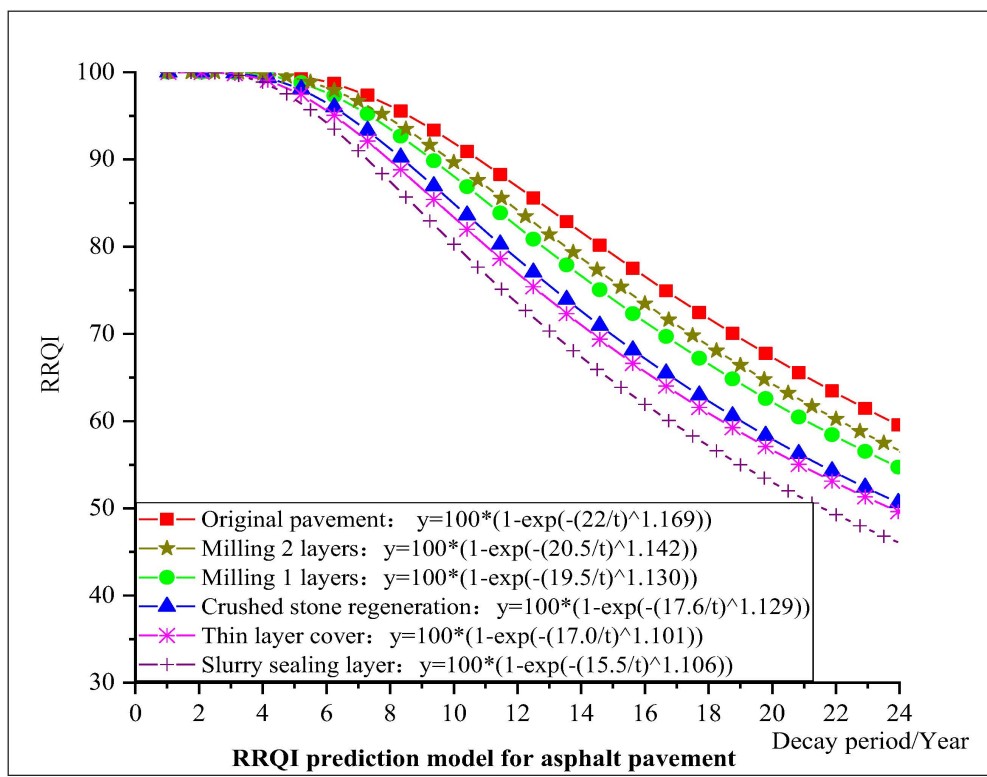

**Figure 6.** RRQI decay curve of asphalt pavement in rural roads over time.

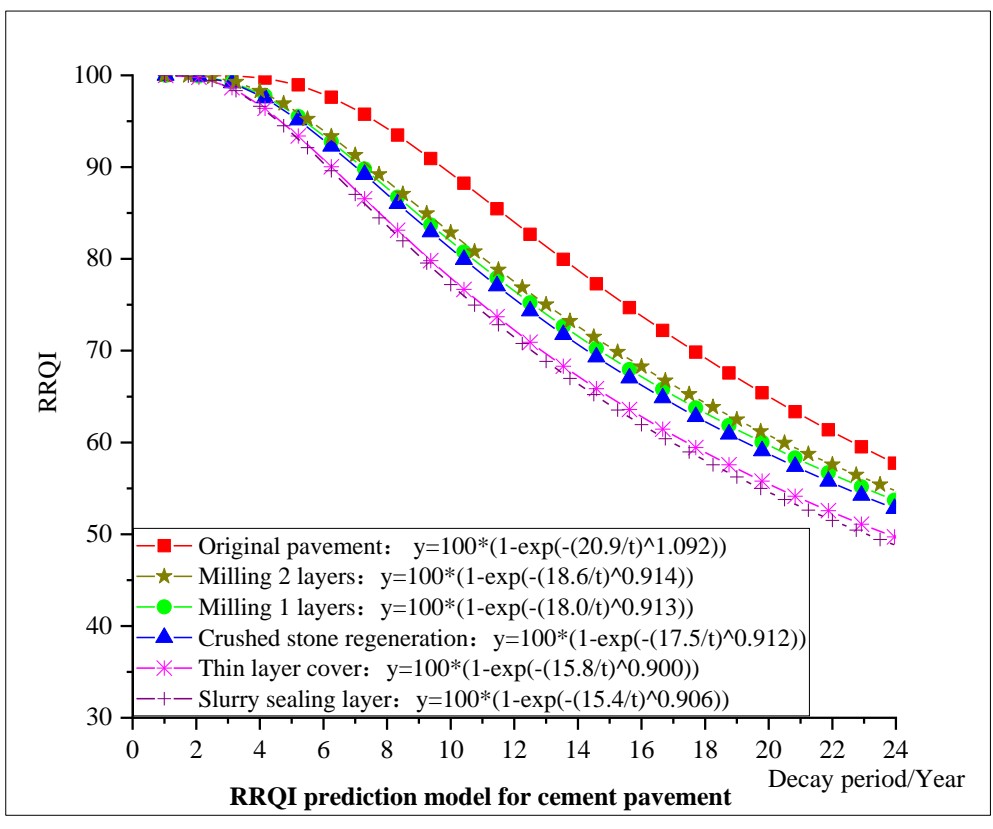

**Figure 7.** RRQI decay curve of cement pavement in rural roads over time.

## 2.3. Optimization Methods

### 2.3.1. Decision Optimization Method

In the rural road network maintenance decision-making process, limited budget, complexity, insufficient program efficiency, and low optimization efficiency are common problems (Chen, W [20] et al. 2021). This paper introduces an improved heuristic optimization algorithm to solve these issues. The method is based on changes in pavement performance after maintenance measures are implemented, aiming to ensure optimal decision-making benefits throughout the life cycle (Chu, J. C [21] et al. 2018). By using a set decision-making model and benefit model, this approach determines the combination of maintenance measures that provides the most long-term benefits for each road section, optimizing a decision-making scheme strategy with the best benefits quickly, as shown in Figure 8. The solution process involves (1) determining the maintenance decision plan for each individual maintenance section within the road network one by one, (2)–(3) combining all maintenance road section decision plans to achieve all possible combinations of programs for the entire road network. (4) The optimization process is based on annual budget cost and a pavement performance-based benefit model as the main optimization link. (5) It calculates and determines the benefits and costs of each maintenance measure based on the benefits model, and (6) conducts a comparative analysis of the cumulative benefits of all strategy schemes through optimization algorithms to determine the best maintenance strategy.

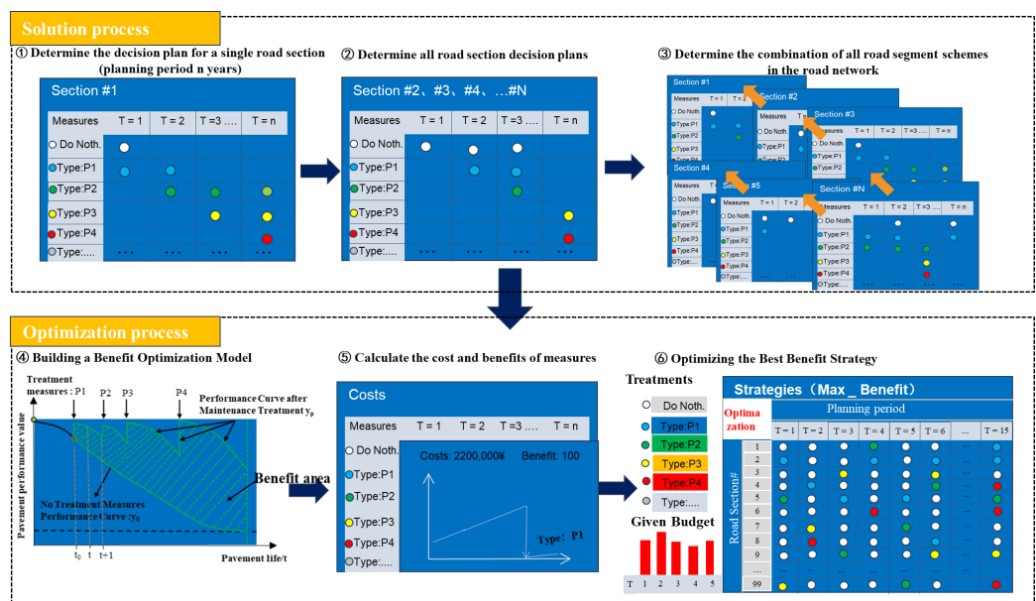

**Figure 8.** Decision steps for an improved heuristic optimization method.

### 2.3.2. Benefit Optimization Calculation

Based on the combination of all road section schemes, the maintenance decision-making optimization work is carried out to determine the best maintenance benefit scheme. The research results define the maintenance benefit area as the area enclosed by the pavement performance curve $y_P$ after treatment measures and the natural attenuation curve y0 of pavement performance without measures, as shown in Figure 9. The Bene_cost ratio of benefit area to its cost is then taken as the final benefit evaluation value of rural road maintenance.

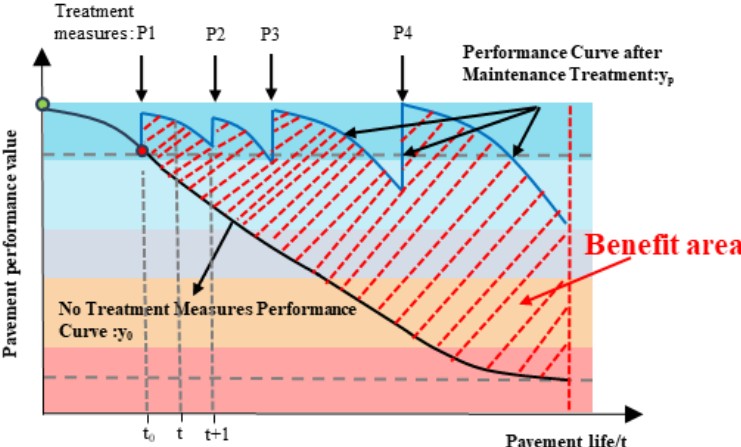

**Figure 9.** Schematic diagram of maintenance benefit area.

Considering the performance decay characteristics and benefit influencing factors of rural road pavement, the *Bene_cost* generated by RPCI and RRQI key performance indicators was calculated, respectively. According to the actual impact degree of each index, it is weighted and summed according to the given weights of 0.65 and 0.35, and the calculation formula of the comprehensive *Bene_cost* based on moment t is constructed, as shown in Equations (4)–(6) below.

$$COST = L * W * UC \tag{4}$$

$$Bene\_cost_t^i = \begin{cases} \sum\limits_{l=1}^{n_l} (\int_{t_0}^{t} y_p^l - y_0^l) / \sum\limits_{l=1}^{n_l} cost_l & \sum\limits_{l=1}^{n_l} cost_l \neq 0 \\ 0 & \sum\limits_{l=1}^{n_l} cost_l = 0 \end{cases} \tag{5}$$

$$Bene\_cost_t = 0.65 * Bene_{cost\,t}(RPCI) + 0.35 * Bene_{cost\,t}(PRQI) \tag{6}$$

where *COST* is the cost of maintenance; *L* and *W* are length and width, respectively; *UC* is the unit area cost (¥/m$^2$); *Bene_cost* is the maintenance benefit calculated based on performance index *i* for the T year; *l* is the lane number; $n_l$ is the number of lanes; $y_p^l$ and $y_0^l$ are the performance curves corresponding to the *l* lane with or without treatment measures taken, respectively.

To improve the calculation efficiency in the process of solution optimization, the improved heuristic optimization method proposed in this paper introduces the allowable deviation variable parameter *D* based on a given budget, and appropriately "relaxes" the cost constraints, as shown in the following formulas (7) and (8), to quickly realize the scheme combination with the best benefit in the whole life cycle.

$$B_t - D_t \leq Cost_t = \sum_{i'=1}^{m} Cost_{i't} \leq B_t + D_t, \; t = 1, \dots, T \tag{7}$$

$$Bene\_cost = \sum_{i'=1}^{m} \sum_{t=1}^{T} Bene\_cost_{i't} \tag{8}$$

where *m* is the number of road sections; $B_t$ is the budget for each planning year; $D_t$ is the maximum allowable deviation from the budget for each planning year.

### 2.4. Data Preparation of Case Studies

Based on the above research results, this paper takes the typical rural road network in Haimen City, China as an example, and carries out medium and long-term (2023–2036) maintenance planning analysis and research. The road network in this case mainly includes

10 routes, including Rui Min Line, Huo Si Line, Guo Xin Line, S336 San He Duan, Yang Hai Line, Hai Tian Line, S336 San Chang Duan, De Hai Line, Dong Tong Line and Shu Gang Line, with a total length of 280 km and two lanes in both directions (the width of one lane is 3.75 m). The original road condition data for this case comes from the actual test data of the rural road network RPCI, RRQI, and the Rural Road Inner Pavement Condition Index (RIPCI) in Haimen City in March 2022. As shown in Table 4, the pavement performance data of some sections are given.

**Table 4.** Performance data of typical sections of Haimen rural road.

| Route | Lane | Starting Station | Ending Station | RPCI | RRQI | RIPCI | Pavement Type |
|---|---|---|---|---|---|---|---|
| Rui Min Line | R (1) | K0 + 000 | K1 + 000 | 96.99 | 95.89 | 98.32 | Asphalt |
| Rui Min Line | R (1) | K1 + 000 | K2 + 000 | 96.75 | 96.21 | 95.15 | Asphalt |
| Rui Min Line | R (1) | K2 + 000 | K3 + 000 | 89.22 | 91.28 | 93.65 | Asphalt |
| …..… | …..… | …..… | …..… | …..… | …..… | …..… | …..… |
| Rui Min Line | R (1) | K19 + 000 | K19 + 950 | 90.34 | 91.12 | 93.80 | Asphalt |
| Huo Si Line | L (1) | K0 + 000 | K1 + 000 | 91.60 | 89.93 | 92.85 | Cement |
| Huo Si Line | L (1) | K1 + 000 | K2 + 000 | 94.64 | 95.20 | 93.46 | Cement |
| Huo Si Line | L (1) | K2 + 000 | K3 + 000 | 95.13 | 96.40 | 95.21 | Asphalt |
| …..… | …..… | …..… | …..… | …..… | …..… | …..… | …..… |
| Guo Xin Line | R (1) | K0 + 000 | K1 + 000 | 92.89 | 93.71 | 95.78 | Asphalt |
| Guo Xin Line | R (1) | K1 + 000 | K2 + 000 | 84.26 | 89.79 | 95.85 | Asphalt |
| Guo Xin Line | R (1) | K2 + 000 | K3 + 000 | 96.39 | 93.74 | 98.45 | Asphalt |
| …..… | …..… | …..… | …..… | …..… | …..… | …..… | …..… |
| YangHai Line | R (1) | K0 + 000 | K1 + 000 | 89.41 | 90.76 | 95.32 | Asphalt |
| YangHai Line | R (1) | K1 + 000 | K2 + 000 | 91.93 | 93.34 | 94.00 | Asphalt |
| …..… | …..… | …..… | …..… | …..… | …..… | …..… | …..… |
| YangHai Line | R (1) | K32 + 000 | K32 + 160 | 91.92 | 92.19 | 96.26 | Asphalt |

At the same time, the road performance data are divided into several standard treatment sections with a length of no more than 1000 m according to the above-mentioned rural road performance evaluation grades, as shown in Figure 10.

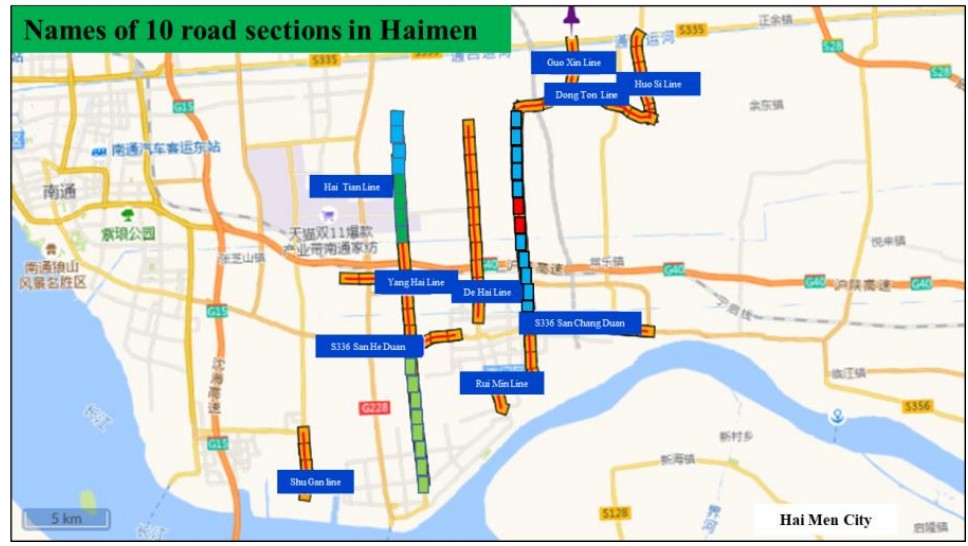

**Figure 10.** Planning section of the rural road network in Haimen City.

## 3. Results and Discussion

Based on the above case data, the budget for different years is calculated by the developed calculation program (the algorithm is realized by system programming), and the results are compared and analyzed to determine the functional maintenance strategy with a reasonable budget and best benefit.

### 3.1. Road Condition Prediction Analysis

According to the existing rural road conditions in Haimen City, the performance requirements in the planning period (the overall average value is not less than 85), and the input costs in previous years. This paper preliminarily selects five annual budget options for decision-making calculation, mainly including plan 1: no budget; plan 2: 5 million (¥); plan 3: 10 million (¥); plan 4: 15 million (¥); plan 5: 5 million (¥); (previous four years: 2023–2026), 10 million (¥) (medium-term five years: 2027–2031), 15 million (¥) (later five years: 2032–2036). As shown in Figures 11 and 12 below, the results of the RPCI grade frequency distribution and predicted value of road surface performance under the optimal strategy of different budget schemes are generated.

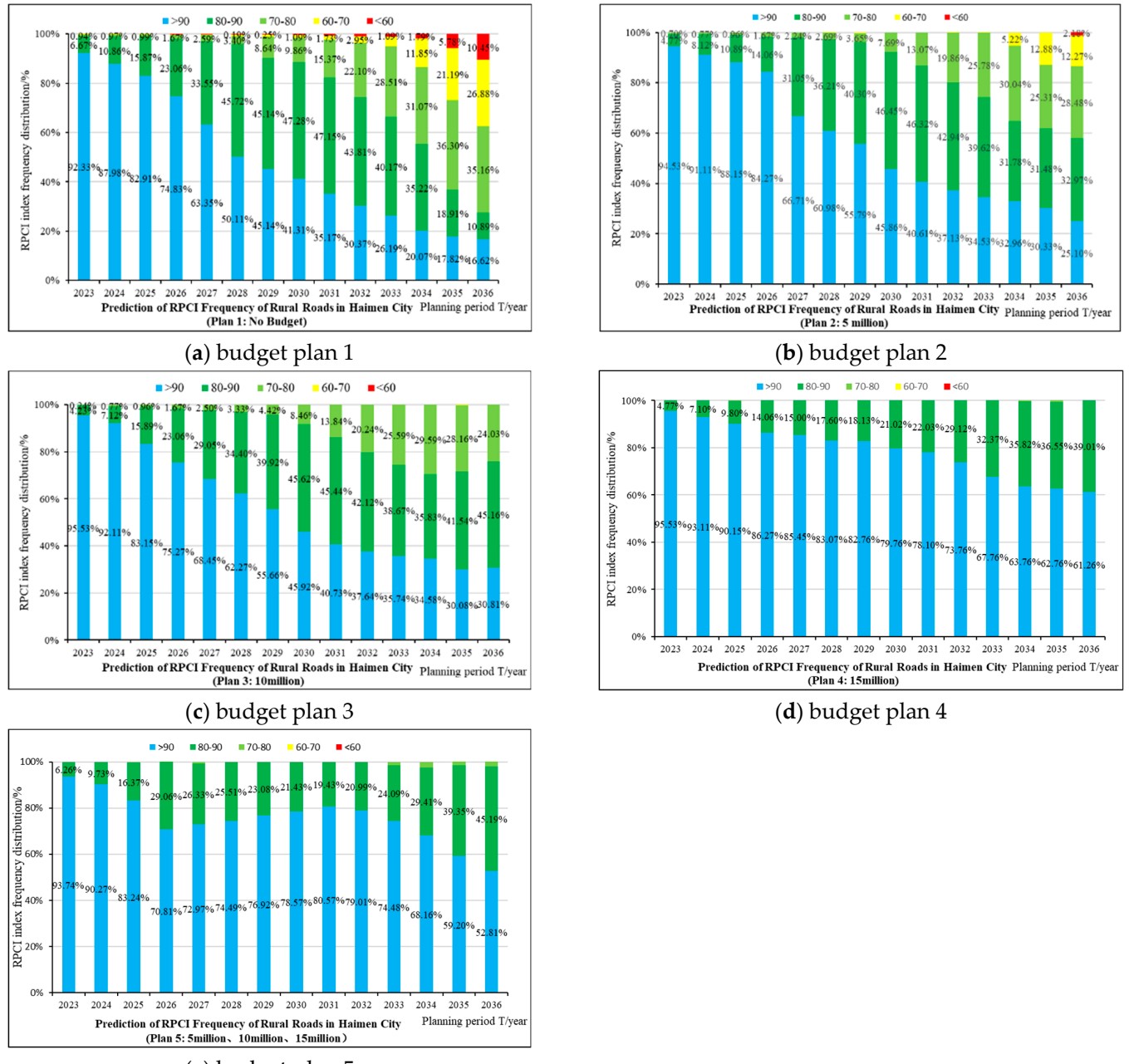

**Figure 11.** Frequency distribution of RPCI on rural roads in Haimen City under different budget plans.

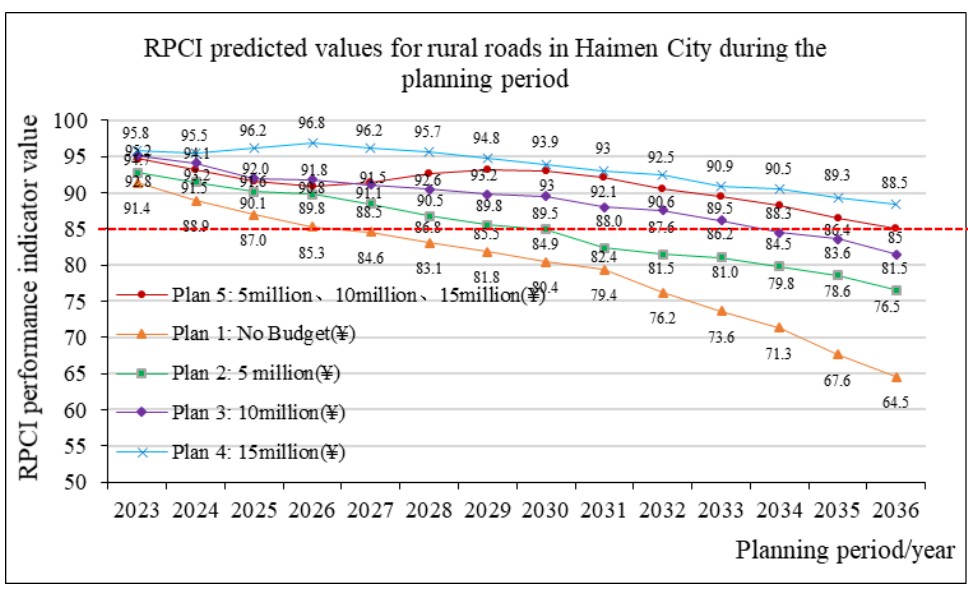

**Figure 12.** Predicted RPCI of rural roads in Haimen City under different budget plans.

At the same time, this paper further predicts and analyzes the pavement RRQI index under different given annual budget plans. Figures 13 and 14 below show the statistical results of RRQI level frequency distribution and performance value in the next 14 years.

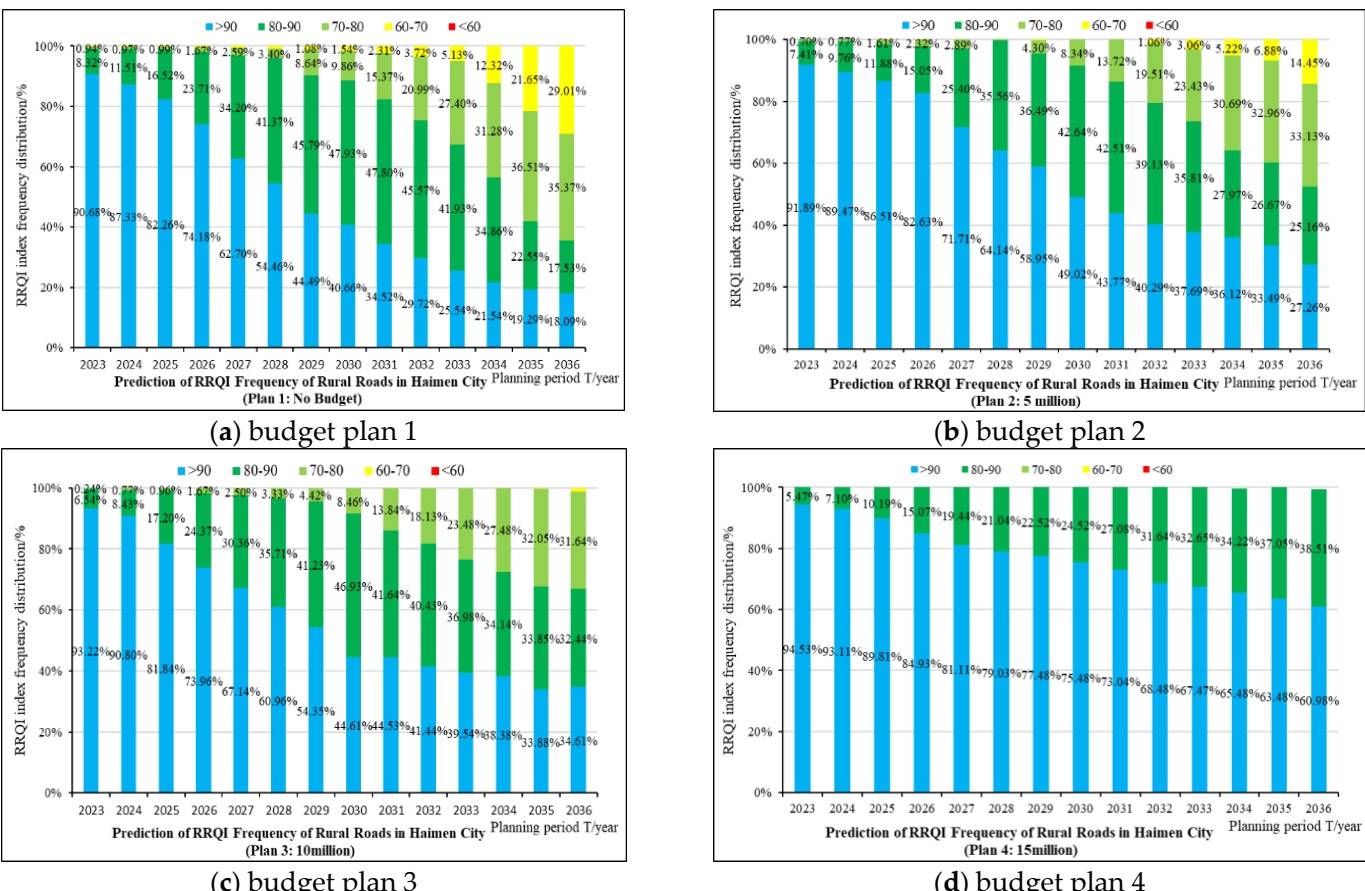

**Figure 13.** *Cont.*

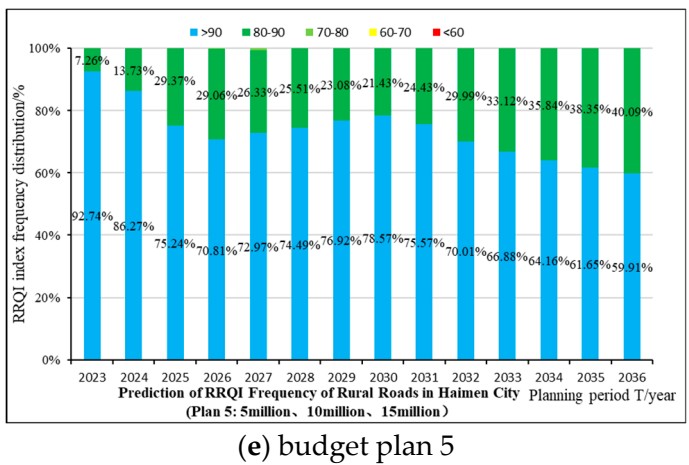

(**e**) budget plan 5

**Figure 13.** Frequency distribution of RRQI on rural roads in Haimen City under different budget plans.

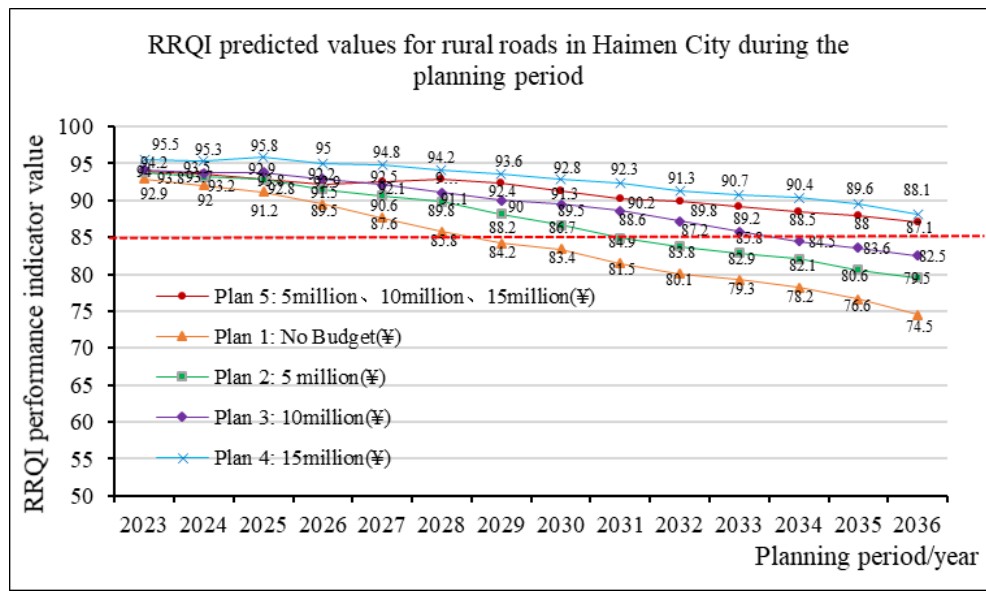

**Figure 14.** Predicted RRQI of rural roads in Haimen City under different budget plans.

According to the above figure, the RPCI and RRQI values of rural road network performance in Haimen City are in a decreasing trend with the increase in the service life of the pavement. By comparing different plans, it is found that with the increase in annual budget investment, the overall performance of the road network gradually improves, and the improvement effect of road surface RPCI compared with RRQI during the planning period is more obvious. When the annual budget reached more than 10 million, the excellent, good, and average grade rate of RPCI and RRQI indicators reached 95%, and the predicted average reached more than 80 points. After adopting the budget investment of plan 5 phased maintenance, its overall road condition performance is at excellent and good levels, and the predicted value is above 85 points.

Therefore, the phased increase in annual budget input according to budget plan 5 is more effective and reasonable than the above-mentioned conservation budget plan. Moreover, the functional priority maintenance strategy has played a significant role in improving the performance of the pavement, which can meet the service level requirements of the municipal road management department for the road network under its jurisdiction during the planning period.

### 3.2. Optimization Scheme Determination

Based on the analysis results of the prediction effect of different budget plans, this paper selects the limited budget plan 5 as the minimum cost required to meet the maintenance performance requirements of the road network in Haimen City and determines the treatment plan that meets the most benefits in the next 14-year life cycle of the planned road section. As shown in Table 5 below, 171 maintenance sections require maintenance measures during the entire road network planning period (treatment schemes for some sections are given).

**Table 5.** Treatment scheme for some sections of maintenance planning in Haimen City.

| Year | Route | Lane | Starting Station | Ending Station | Treatment Measures | Cost (China RMB)/¥ | Bene_ Cost |
|---|---|---|---|---|---|---|---|
| 2023 | Guo Xin Line | L (1) | K1 + 000 | K2 + 000 | Milling and repaving 1 layer | 502,500 | 2.82 |
| 2023 | Dong Tong | L (1) | K3 + 000 | K4 + 000 | Crushed stone regeneration | 270,000 | 3.77 |
| 2023 | Yang Hai Line | R (1) | K17 + 000 | K18 + 000 | Slurry sealing layer | 131,250 | 9.14 |
| 2023 | Guo Xin Line | R (1) | K1 + 000 | K2 + 000 | Milling and repaving 1 layer | 502,500 | 2.82 |
| 2023 | Rui Min Line | R (1) | K7 + 000 | K8 + 000 | Milling and repaving 1 layer | 502,500 | 2.86 |
| 2023 | Yang Hai Line | R (1) | K5 + 000 | K6 + 000 | Milling and repaving 1 layer | 502,500 | 2.90 |
| 2023 | Rui Min Line | L (1) | K2 + 000 | K3 + 000 | Milling and repaving 1 layer | 502,500 | 2.92 |
| 2023 | Rui Min Line | R (1) | K8 + 000 | K9 + 000 | Milling and repaving 1 layer | 502,500 | 2.86 |
| 2024 | Rui Min Line | L (1) | K4 + 000 | K5 + 000 | Milling and repaving 1 layer | 502,500 | 3.36 |
| 2024 | Yang Hai Line | R (1) | K20 + 000 | K21 + 000 | Crushed stone regeneration | 270,000 | 4.44 |
| 2024 | Yang Hai Line | L (1) | K13 + 000 | K14 + 000 | Crushed stone regeneration | 270,000 | 4.44 |
| 2025 | Yang Hai Line | R (1) | K8 + 000 | K9 + 000 | Milling and repaving 2 layer | 502,500 | 3.54 |
| 2025 | Rui Min Line | L (1) | K15 + 000 | K16 + 000 | Milling and repaving 2 layer | 502,500 | 3.74 |
| 2025 | Guo Xin Line | L (1) | K0 + 000 | K1 + 000 | Milling and repaving 2 layer | 502,500 | 3.54 |
| ...... | ...... | ...... | ...... | ...... | ...... | ...... | ...... |
| 2036 | Rui Min Line | L (1) | K5 + 000 | K6 + 000 | Milling and repaving 2 layer | 1,117,500 | 0.22 |
| 2036 | Rui Min Line | R (1) | K2 + 000 | K3 + 000 | Milling and repaving 2 layer | 1,117,500 | 0.23 |

Through the statistics of the above-mentioned maintenance planning results of Haimen City, Figures 15 and 16 give the area and maintenance cost, respectively, of the types of treatment measures taken each year in the whole road network planning period.

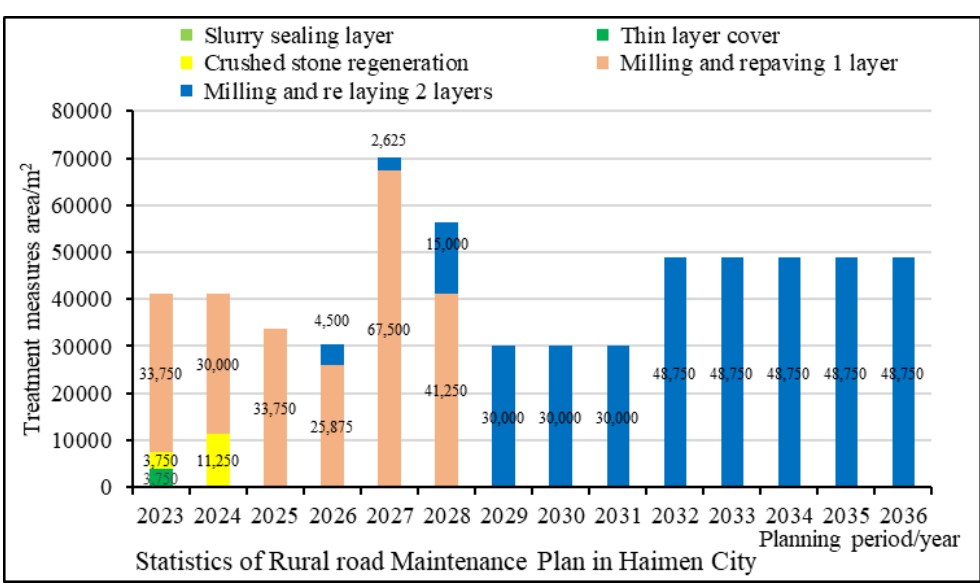

**Figure 15.** Area statistics of treatment measures in the Haimen rural road planning period.

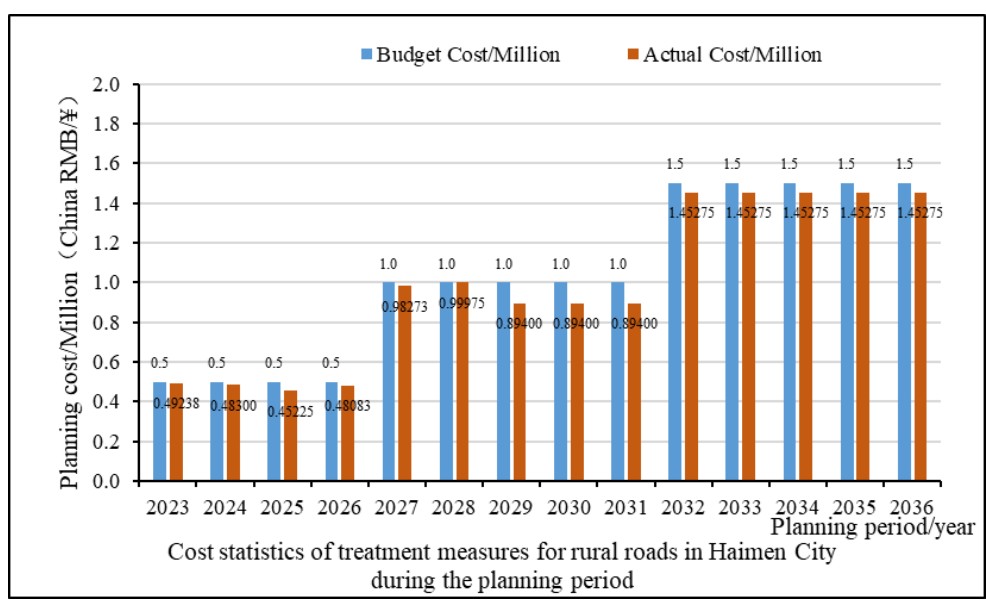

**Figure 16.** Statistics of treatment measures during the planning period of a rural road in Haimen City.

According to the planning scheme and forecast results determined in the above figure, it can be seen that the overall road condition of the Haimen rural road network is good. In the next two years, the maintenance plan will mainly adopt various treatment measures such as a slurry sealing layer, crushed stone regeneration, and milling and repaving one layer. With the increase in pavement service life, pavement performance will be degraded to a certain extent. In view of the possible local typical diseased sections of the pavement, the planning and decision-making results for 2025–2028 mainly adopt milling and repaving one layer to improve pavement performance. Finally, in the middle and late stages of the planning (2029–2036), pavement performance will be greatly reduced, and the decision-making result is that milling and repaving two layers of strong maintenance measures are adopted to maintain the pavement's service level.

## 4. Conclusions

This paper analyzes the current situation of complex rural road networks, large maintenance scales, and limited budgets in China. It explores how to conduct medium and long-term scientific maintenance planning to achieve the best maintenance strategy within the service life cycle. The study concludes:

(1) Based on the maintenance characteristics of rural roads in China, typical maintenance technologies suitable for different strength grades are analyzed and selected. RPCI and RRQI decision tree models for asphalt and cement pavements are established, proposing maintenance countermeasure sets under different performance combinations.

(2) Considering the impact of pavement structure, maintenance history, and traffic volume on performance degradation, typical rural road sections in cities and counties like Haimen, Guannan, and Yangzhou in Jiangsu province were selected. By fitting and analyzing a large amount of detection data, RPCI and RRQI pavement performance prediction models based on five treatment grades are established.

(3) To address the complex solution process of rural road maintenance decision-making, an improved heuristic optimization method is proposed, establishing a model based on pavement performance benefit. Through optimization calculations, a maintenance strategy with the best benefits in the life cycle is quickly generated.

(4) A case study of ten typical rural road sections in Haimen City, Jiangsu province was conducted to apply long-term maintenance planning from 2023 to 2036. By comparing and analyzing the prediction effects of pavement performance (RPCI and RRQI) under different budget plans, a maintenance strategy with a reasonable budget



and maximum benefit for the planned road section in the next 14 years life cycle is determined, verifying the feasibility and effectiveness of the research model.

**Author Contributions:** Conceptualization, C.H.; methodology, C.H.; software, J.H.; validation, X.Y.; investigation, C.H.; data curation, L.C. and T.C.; writing—original draft preparation, C.H. and J.H; writing—review and editing, C.H.; visualization, C.H.; funding acquisition, C.H. All authors have read and agreed to the published version of the manuscript.

**Funding:** This work was supported by the Jiangsu Science and Technology Innovation Support Program of China (No.BZ2022019), Nanjing Science and Technology Program (No.2021-12004), and the National Key Research and Development Program of China (No.2020YFAO714302).

**Institutional Review Board Statement:** Not applicable.

**Informed Consent Statement:** Not applicable.

**Data Availability Statement:** The data presented in this study are available on request from the corresponding author. (Some data are related to research projects, and are not provided for the time being).

**Conflicts of Interest:** The authors declare no potential conflict of interest.

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
