# Peer review of "Long-Term Maintenance Planning Method of Rural Roads under Limited Budget: A Case Study of Road Network"

_applsci, doi:10.3390/app132312661_

Round 1

Reviewer 1 Report

Comments and Suggestions for Authors

Long-term maintenance planning method of rural roads under limited budget: A case study of road network

Comments:

This paper shows an interesting topic on proposing a long-term maintenance planning method of rural roads. A minor revision is needed before publication. Some suggestions are provided as follows:

(1) Abstract: Some quantitative conclusions are expected here,

(2) Introduction: Make the research gap and objective clearer. Some reasons for road damage can be added, such as aging and heavy loading. See references: Towards a sustainable optimization of pavement maintenance programs under budgetary restrictions. Journal of Cleaner Production. 2017. Toward the long-term aging influence and novel reaction kinetics models of bitumen. International Journal of Pavement Engineering. 2022.

(3) The font format in Tables should be the same as the text.

(4) The quality of figures should be improved, for instance: the legend in Figures 4 and 5 are hard to review.

(5) Please enlarge the figure size of figures 11 and 13.

(6) The error bar can be added in figures.

(7) Some recommendations for future work can be provided.

(8) Some practical pictures of damage rural roads can be added.

Comments on the Quality of English Language

Check and revise the English editing of this manuscript, please.

Reviewer 2 Report

Comments and Suggestions for Authors

- In the second chapter (Research Conditions and Methods) in the subsection (2.1.Key indicators and conditions) in point (2.1.1. Pavement performance evaluation) in line 97, the full meaning of the abbreviation RPCI must be stated, and in line 98 the full meaning must be stated abbreviations RRQI.

- In Table 1, it is necessary to indicate next to IRI (m/km), and it is necessary to indicate measurement units next to CR, as well as for RPCI or RRQI.

- In the text before Table 2, it is necessary to clarify what is meant by P3 (crushed stone regeneration)?

- In Tables 2 and 7 should the prices be expressed in pounds or dollars and clarify the meaning of (pci_pms+10) found in Table 2?

- Figure 1 shows the Milling and repaving 3 layers option, and Table 2 does not show that rural road option, should it be P6?

- In point 2.2.1. Performance decay analysis in the text before Figure 3 should be bold curve a, curve b and curve c.

- The legend related to the curves of the graph in Figures 4, 5, 6 and 7 should be enlarged to be readable.

- Figure 6 is numbered twice, it should be changed to Figure 7 (line 216).

- In subsection 2.4. Data Preparation of Case Studies in line 280 it is necessary to state the full meaning of the abbreviation RIPCI?

- Figures 12, 13, 15 and 16 should be enlarged to make them clearer.

- In the title of Figures 15 (line 356) and 16 (line 358), Table should be removed.

Reviewer 3 Report

Comments and Suggestions for Authors

This paper provides a case study of long-term maintenance planning method of rural roads. A pavement performance prediction model and a heuristic optimization are proposed and applied in this case study. It is well written, and I have some minor comments to improve the quality and emphasize the impact of this paper:

1. Figure 4 and Figure 5 shows the RPCI decay curve of asphalt pavement and cement pavement in rural roads over time. The same information is shown in Table 4 to describe the relationship between treatment measures and decay curve of RPCI. It is suggested to remove one of them to avoid redundant information.

2. Same for Figure 4/5 and Table 5.

3. What is the unit of cost used in this paper? It is suggested to label it explicitly in the figures if all the costs are using Chinese currency.

4. This research used RPCI and RRQI as pavement performance indicators and correlated them with maintenance cost in Equation 5 to Equation 7. However, recently there are many papers showing that IRI (international roughness index) has more direct effect on the ride quality, fuel consumption, and maintenance cost. For example, paper “Mechanistic Excess Fuel Consumption of a 3D Passenger Vehicle on Rough Pavements” investigated the excess fuel consumption and developed an ERSI model to estimate the fuel consumption based on pavement surface condition. Can you discuss this paper in your research? This could help emphasize the impact of your research.

Reviewer 4 Report

Comments and Suggestions for Authors Maybe it would be necessary to develop the introduction part. Comments on the Quality of English Language Minor grammatical agreements.

Author Response

Thank you for pointing out this point. We agree with your opinion. We have made corrections to grammar errors and some parts of the English expression throughout the text.